# A Quantitative Assay for Ca^2+^ Uptake through Normal and Pathological Hemichannels

**DOI:** 10.3390/ijms23137337

**Published:** 2022-06-30

**Authors:** Chiara Nardin, Abraham Tettey-Matey, Viola Donati, Daniela Marazziti, Chiara Di Pietro, Chiara Peres, Marcello Raspa, Francesco Zonta, Guang Yang, Maryna Gorelik, Serena Singh, Lia Cardarelli, Sachdev S. Sidhu, Fabio Mammano

**Affiliations:** 1CNR Institute of Biochemistry and Cell Biology, Monterotondo, 00015 Rome, Italy; chiara.nardin@iit.it (C.N.); abraham.matey@ibbc.cnr.it (A.T.-M.); viola.donati@ibbc.cnr.it (V.D.); daniela.marazziti@cnr.it (D.M.); chiara.dipietro@cnr.it (C.D.P.); chiara.peres@ibbc.cnr.it (C.P.); marcello.raspa@cnr.it (M.R.); 2Shanghai Institute for Advanced Immunochemical Studies, ShanghaiTech University, Shanghai 201210, China; fzonta@shanghaitech.edu.cn (F.Z.); yangguang@shanghaitech.edu.cn (G.Y.); 3The Donnelly Centre, University of Toronto, Toronto, ON M5S 3E1, Canada; maryna.gorelik@utoronto.ca (M.G.); singhserena@gmail.com (S.S.); lia.cardarelli@utoronto.ca (L.C.); 4Department of Physics and Astronomy “G. Galilei”, University of Padua, 35131 Padua, Italy

**Keywords:** connexins, genetically encoded calcium indicators, monoclonal antibodies, drug discovery, genodermatoses, cancer, lentivirus, bicistronic vectors

## Abstract

Connexin (Cx) hemichannels (HCs) are large pore hexameric structures that allow the exchange of ions, metabolites and a variety of other molecules between the cell cytoplasm and extracellular milieu. HC inhibitors are attracting growing interest as drug candidates because deregulated fluxes through HCs have been implicated in a plethora of genetic conditions and other diseases. HC activity has been mainly investigated by electrophysiological methods and/or using HC-permeable dye uptake measurements. Here, we present an all-optical assay based on fluorometric measurements of ionized calcium (Ca^2+^) uptake with a Ca^2+^-selective genetically encoded indicator (GCaMP6s) that permits the optical tracking of cytosolic Ca^2+^ concentration ([Ca^2+^]_cyt_) changes with high sensitivity. We exemplify use of the assay in stable pools of HaCaT cells overexpressing human Cx26, Cx46, or the pathological mutant Cx26G45E, under control of a tetracycline (Tet) responsive element (TRE) promoter (Tet-on). We demonstrate the usefulness of the assay for the characterization of new monoclonal antibodies (mAbs) targeting the extracellular domain of the HCs. Although we developed the assay on a spinning disk confocal fluorescence microscope, the same methodology can be extended seamlessly to high-throughput high-content platforms to screen other kinds of inhibitors and/or to probe HCs expressed in primary cells and microtissues.

## 1. Introduction

Connexin (Cx) hemichannels (HCs) may function as large-pore plasma membrane channels or dock head-to-head with other HCs from an opposing cell, leading to the formation of intercellular gap junction channels (IGJCs) [1,2,3,4]. Both Cx HCs and IGJCs are permeable to a variety of molecular species, such as ions, second messengers, metabolites and even small noncoding RNA (siRNAs and miRNAs). Based on pore size estimates, the list of permeating molecules could potentially include >35,000 members of the Human Metabolome Database [5], including Ca^2+^ [6,7,8,9,10].

Not only Ca^2+^ can permeate through Cx HCs but, in addition, HC gating depends on Ca^2+^ concentration in the extracellular milieu ([Ca^2+^]_ex_). In particular, it is well known that lowering the [Ca^2+^]_ex_ favors the opening of HCs formed by several different types of Cxs, whereas relatively high Ca^2+^ concentrations prevent opening [11,12,13,14,15,16,17]. Numerous other factors modulate the open probability of Cx HCs, including mechanical strain, changes in pH, transmembrane voltage, post translational modifications (especially protein phosphorylation), cellular redox state, metabolic inhibition, nitric oxide and linoleic acid; reviewed in [17,18,19,20]. 

The opening of Cx HCs promotes the diffusive release of paracrine messengers, including ATP [21,22,23,24,25,26], glutamate [27], prostaglandins [28], NAD(+) [29] and glutathione [30,31]; reviewed in [32]. In several instances, molecules released through open Cx HCs, most importantly ATP, may act as damage-associated molecular patterns (DAMPs) [33], activating inflammatory pathways and promoting cell death [34,35,36,37]. Not surprisingly, point mutations that render Cx HCs constitutively more active or abnormally active (referred to also as “leaky” HCs) have been implicated in numerous genetic diseases affecting the facial appearance, dentition (small and carious teeth), eyes (microphthalmia, microcornea), fingers (syndactyly), lens (cataracts), nervous system (myelination defects), heart (atrial fibrillation), inner ear (deafness) and skin (various disorders); reviewed in [38,39,40]. 

Deregulated Ca^2+^ flux through HCs has been implicated in several diseases linked to Cx mutations [18,38,39,40]. In addition, the opening of wild type (wt) Cx HCs is associated with an impressive list of pathological conditions that affect a major proportion of the world population, including ischemia/stroke, Alzheimer’s disease, epilepsy, liver fibrosis and cirrhosis, nonalcoholic steatohepatitis, inflammation and others [41,42,43,44,45,46]. Of note, inhibition of Cx HCs in animal models in vivo exerted anticonvulsant effects [47], lowered the degree of liver fibrosis induced by administration of thioacetamide [48] and improved recovery from spinal cord injury [49]. 

Cx HC activity has been probed indirectly by the release of ATP and other metabolites as mentioned above [21,22,23,24,25,26]. Other classical methods include electrophysiology (patch clamp) and dye uptake or leakage assays. The latter are based on passive diffusion of fluorescent molecules through open HCs in the cell plasma membrane [50]. Experiments of this type have been mainly conducted in model cell systems but occasionally also in native tissues [24,25] or in vivo [26]. The aim of this work was to establish a complementary, all-optical method to investigate Cx HC activity based on the uptake of Ca^2+^. 

We focused on Cx26, which is expressed at extremely low, oftentimes undetectable levels in epidermal keratinocytes, yet, paradoxically, mutant Cx26 leaky/hyperactive HCs cause genodermatoses, including keratitis-ichthyosis-deafness syndrome (KID) and palmoplantar keratoderma with deafness (PPKDFN) that may drastically compromise quality of life and life-long health [40,51,52] (Table 1). To broaden the impact of this study and demonstrate the wider applicability of the methodology, we also investigated HCs composed of Cx46 [11,12], which is an important target for cancer therapy [53,54,55,56,57] and, when mutated, is involved in cataract formation [58]. 

The Ca^2+^ uptake assay we developed provides a highly sensitive readout of Cx HC activity, as detailed hereafter. Furthermore, the assay is also suitable to evaluate the efficacy of different HC inhibitors that are attracting a growing attention as potential therapeutics [41,42,43,44,45,46,47,48], including antibodies targeting the extracellular domain of HCs [49,59,60,61,62].

## 2. Results

### 2.1. Generation of Stable Cell Pools Overexpressing the Cx of Interest and GCaMP6s 

HaCaT cells are spontaneously immortalized line of aneuploid human keratinocytes [63] widely adopted to model the epidermis in vitro [64,65]. We and others previously overexpressed Cx HCs in HaCaT cells [59,61,66]. Here, given that uncontrolled expression of leaky HCs causes toxicity and rapidly leads to cell death (see, e.g., [66,67,68,69,70,71,72,73]), we designed and produced lentiviruses (LVs) for Tet-on bicistronic expression of the Cx of interest and the genetically encoded indicator GCaMP6s that permits optical tracking of [Ca^2+^]_cyt_ changes with high sensitivity [74] (see Materials and Methods)**.**

We inserted the Cx coding sequence (CDS) upstream of an internal ribosome entry site (IRES) sequence, in turn, followed by the CDS of GCaMP6s. Next, by puromycin selection of virally transduced cells, we obtained stable cell pools expressing human Cx26 (referred to as HaCaT-Cx26-GCaMP6s cells) or the lethal KID-related mutant Cx26G45E that generates leaky HCs [68,70,75,76,77] (referred to as HaCaT-Cx26G45E-GCaMP6s cells). The expression levels of both IRES and GCaMP6s mRNAs increased over 350-fold within 24 h of doxycycline (dox) addition (2 µg/mL) to the culture medium (Figure 1). The non-zero levels of both mRNAs in the absence of dox can be attributed to the minimal basal leakiness of the TRE promoter in the lentiviral construct [78].

Analyses of HaCaT-Cx26-GCaMP6s cells by live microscopy showed a monotonic growth of GCaMP6s fluorescence levels with time, reaching a steady state after ~30 h from the onset of dox exposure (Figure 1C). In HaCaT-Cx26-GCaMP6s cells pre-exposed to dox for 24 h, we confirmed overexpression of Cx26 by qPCR (Figure 2A) as well as by immunofluorescence staining with a commercial antibody that recognizes an intracellular epitope (Figure 2B). Note that Cx26 was detected at very low levels also in the absence of dox. Immunoreactivity against Cx26 produced fluorescent puncta at points of contacts between adjacent cells, corresponding to IGJCs (Figure 2C).

To confirm that functional HCs were present in the plasma membrane of live HaCaT-Cx26-GCaMP6s cells, we performed a standard dye uptake assay (Figure 3) using 4′,6-diamidino-2-phenylindole (DAPI, 5 µM) that is non fluorescent in the medium but becomes fluorescent upon penetrating into the cell and binding to nucleic acids [50]. 

Cells pre-exposed to dox for 24 h showed a significant increase of DAPI fluorescence emission when bathed in an extracellular medium (ECM) containing a low [Ca^2+^]_ex_ (60 µM). HaCaT cells and isolated primary keratinocytes can be cultured in this and even lower [Ca^2+^]_ex_ values [63,64,65,80,81,82] (which increase the open probability of HCs [11,12,13,14,15,16,17,18,19,20]). Indeed, the cell line “was designated HaCaT to denote its origin from human adult skin keratinocytes propagated under low Ca^2+^ conditions and elevated temperature” [63]. Supplementing ECM with 2 mM Ca^2+^, which reduces the open probability of HCs [11,12,13,14,15,16,17,18,19,20], inhibited DAPI uptake (Figure 3A). In contrast, in prior work with parental HaCaT cells that are known to express low levels of Cxs [59] and mainly Cx43 [83,84,85], DAPI uptake lacked the sensitivity required to discriminate between low and high [Ca^2+^]_ex_ conditions [59]. 

In the experiments of Figure 3, DAPI uptake was also inhibited by flufenamic acid (FFA, 100 µM), a well-known and widely used aspecific blocker of HCs [77] (Figure 3B). To corroborate these findings, we used abEC1.1m, a previously characterized antagonist mAb, formatted as a single chain fragment variable (scFv)-fragment constant (Fc) diabody [59] with human scFv and mouse Fc domain, which specifically binds to the extracellular domain of Cx26, Cx30 and Cx32 HCs, however, is not effective on Cx43 HCs [60]. The effect of abEC1.1m (1 μM) was like that of FFA (Figure 3B). The simplest explanation for these results is that DAPI uptake occurred primarily through Cx26 HCs, whose contribution prevailed over that of HCs formed by other residual Cx isoforms expressed in dox-induced HaCaT-Cx26-GCaMP6s cells.

### 2.2. Development and Validation of the Ca^2+^ Uptake Assay

Based on the outcome of the experiments described above, we deemed it worthwhile to conduct Ca^2+^ uptake experiments in confluent cultures of HaCaT-Cx26-GCaMP6s cells using a custom-built spinning disk confocal fluoresce microscope (described in [86]) equipped with a 20× water-immersion objective. The wide field of view (FOV) achieved with this objective (630 µm ∅) allowed us to collect and average images of GCaMP6s fluorescence emission from more than 100 cells per culture. Experiments were carried out in accordance with the following procedure (Appendix A).

First, we pre-incubated the cells under the microscope in ECM for about 15 min to allow [Ca^2+^]_cyt_ to equilibrate with the low [Ca^2+^]_ex_ (60 µM) of the medium. Next, we excited GCaMP6s using a 488 nm laser and started to collect fluorescence images at 1 frame/s for 20 s of baseline (pre-stimulus) recording. Then, we added a bolus of CaCl_2_ (2 µL) to the medium in the recording chamber (1 mL volume) at different concentrations in different experiments, aiming to achieve final estimated [Ca^2+^]_ex_ of 100 µM, 260 µM, 560 µM, 1 mM and 2 mM and continued to acquire images at 1 frame/s for a total of 10 min. By the end of each recording interval, we applied a bolus of the Ca^2+^ ionophore ionomycin (200 µM, 2 µL and 0.4 µM estimated final concentration) to confirm that GCaMP6s signals were not saturated (Appendix A). Thereafter, the experiment was terminated, and the culture was discarded.

For each experiment, GCaMP6s fluorescence emission was quantified pixel-by-pixel and frame-by-frame as Δ*F* = *F* − *F*_0_, where *F* represents emission at time *t,* and *F*_0_ represents emission averaged over all baseline (pre-stimulus) frames [87]. We then spatially averaged Δ*F* pixel signals over all cells in the FOV and produced a single Δ*F* trace for each experiment. In the first 20 s after CaCl_2_ addition, HaCaT-Cx26-GCaMP6s pre-exposed to dox for 24 h responded with a transient peak of [Ca^2+^]_cyt_ followed by a delayed and broader secondary maximum the height of which depended on CaCl_2_ bolus concentration (see Discussion) (Figure 4A). Ionomycin generated a third response peak that largely exceeded the two previous ones (Appendix A). 

Cells not exposed to dox showed minimal responses due to the basal leakiness of the TRE promoter (Appendix A). For statistical analyses, we normalized each Δ*F* trace for the peak response to ionomycin, Δ*F*_max_ (Appendix A). We computed average Δ*F*/Δ*F*_max_ traces from *n* independent experiments (see figure legends) in each condition, using a different culture for each experiment and displayed them as the mean (solid line) ± standard error of the mean (s.e.m., dashed lines, Figure 4A).

Finally, we quantified the [Ca^2+^]_cyt_ load in terms of the area subtended by the Δ*F*/Δ*F*_max_ traces. The data analysis showed that the sensitivity of the assay plateaued between 1 mM and 2 mM final [Ca^2+^]_ex_ (Figure 4B), therefore, we adopted 2 mM as the standard final [Ca^2+^]_ex_ for all the other experiments. Reducing the initial [Ca^2+^]_ex_ from 60 µM to 20 µM abolished the delayed and broader secondary maximum, resulting in a dramatically reduced assay sensitivity (Appendix A).

We next tested the effects of FFA and abEC1.1m on Ca^2+^ uptake quantified as explained above. Based on these analyses, abEC1.1m (1 µM, in ECM) exerted an inhibitory effect comparable to that of FFA (100 µM, in ECM; *P* = 1, ANOVA; Figure 4C and Appendix A), suggesting that Cx26 HCs provided the dominant uptake route for Ca^2+^. In this set of experiments, we tested also a new fully human immunoglobulin G (IgG) format of this mAb, henceforth referred as abEC1.1h-IgG (1 µM, in ECM). Although abEC1.1h-IgG modified the kinetics of Ca^2+^ uptake differently from abEC1.1m (Figure 4C), the overall blockade effect exerted by the two mAbs was undistinguishable (*P* = 1, ANOVA; Figure 4D).

To better characterize the new IgG, we constructed a dose–inhibition curve (Figure 4E and Appendix A) and determined that the half-maximal effective concentration (EC_50_) was comprised between 1 and 10 nM. At saturating concentrations (>1 µM), the inhibitory effect of abEC1.1h-IgG was about 80%, comparable to that the ScFv-Fc format previously estimated by the patch-clamp technique [59,61]. Immunolabeling with abEC1.1h-IgG produced fluorescent puncta at cell boundaries but not at points of contact between adjacent cells (Figure 4F; compare to Figure 2C) consistent with binding of abEC1.1h-IgG to unpaired HCs in the cell plasma membrane.

### 2.3. mAbs Targeting the Extracellular Domain of Cx26 Inhibit Ca^2+^ Uptake through Leaky HCs Implicated in KID Syndrome

Having successfully established working conditions to conduct Ca^2+^ uptake assays, we turned to cells overexpressing Cx26G45E. The percentage of pyknotic nuclei in HaCaT-Cx26G45E-GCaMP6s cells exposed to dox for 24 h (Table 2) was not significantly different from that of cultures not exposed to dox (Table 3). Therefore, we used cells pre-exposed to dox for 24 h to compare the effect of ECM alone or ECM supplemented with FFA (100 µM), abEC1.1m (1 µM) or abEC1.1h-IgG (1 µM). All three inhibitors significantly reduced Ca^2+^ uptake (Figure 5), suggesting that Ca^2+^ uptake was mediated by Cx26G45E HCs rather than by passive leakage through a compromised plasma membrane. Based on patch-clamp experiments conducted on Cx26G45E mutant HCs [59] and in vivo experiments on a mouse model of a related genodermatosis [61], these results suggest that the tested mAbs are candidate therapeutics for KID syndrome.

### 2.4. Effects of Uncharacterized mAbs, Targeting the Extracellular Domain of Cx46, on Ca^2+^ and DAPI Uptake

For the final set of experiments, we generated a stable pool of HaCaT-Cx46-GCaMP6s cells and used them to test new uncharacterized mAbs targeting the extracellular domain of Cx46 (Figure 6).

FFA (100 µM) strongly reduced Ca^2+^ uptake in HaCaT-Cx46-GCaMP6s cells pre-exposed to dox for 24 h. In contrast, we observed a significant, clear-cut inhibitory effect of about 63% only with Ab17396 at 10 µM concentration. At two lower concentrations (0.1 and 1.0 µM), we could not estimate the [Ca^2+^]_cyt_ load from the area subtended by the ΔF/ΔF_max_ traces because, surprisingly, the traces dropped below the pre-stimulus levels after CaCl_2_ bolus delivery (see Discussion). 

For a complementary estimate of Ab17396 activity, we performed DAPI uptake experiments in ECM alone or supplemented with Ca^2+^ (2 mM), FFA (100 µM) and Ab17396 (1.0 or 10 µM) (Figure 7). DAPI uptake was significantly inhibited by both Ca^2+^ and FFA, whereas the reductions provided by Ab17396 were not statistically significant (*P* = 0.085 between ECM and 10 µM Ab17396 conditions). However, these results were negatively affected by the limited number of experimental replicates that we were able to collect due to the shortage of Ab17396.

Finally, we also performed DAPI uptake experiments with other anti-Cx46 mAbs from the pool that led to the selection of Ab17396 (Figure 8). Together, based on the results of this preliminary screening, two of them, Ab17396 and Ab17398, appear worthy of further analyses to determine their potency, selectivity and usefulness as Cx46 HC inhibitors.

## 3. Discussion

In this work, we developed a novel all-optical assay to probe Ca^2+^ uptake through normal or pathological HCs, which are emerging drug targets for a variety of pathological conditions [41,42,43,44,45,46,47,48,49,61,62], using cytosolic GCaMP6s fluorescence intensity variations as a readout. This choice was dictated by different considerations. On the one hand, the study of Ca^2+^ dynamics in cells expressing pathological HCs is of utmost importance, given that deregulated Ca^2+^ influx through HCs has been implicated in different disorders, including but not limited to those affecting the skin, such as KID syndrome [40] and Clouston syndrome [61]. 

On the other hand, fluorometric assays based on intracellular Ca^2+^ variations are industry standards widely used in drug screening for membrane channels and G-protein coupled receptors (GPCRs) [88,89]. As shown here, the assay provided a sensitive indication for the effect of HC inhibitors, considering that it allowed us to detect changes in the uptake of Ca^2+^ with as little as 0.1 nM of the abEC1.1h-IgG mAb in HaCaT-Cx26-GCaMP6s cells. 

In these experiments, GCaMP6s signals exhibited complex kinetics with an early peak in close temporal proximity to the CaCl_2_ bolus application, followed by a delayed response that generated a second relative maximum with a delay of 2 to 3 min. The reason for this biphasic response is unclear. We speculate that the sudden increase of the [Ca^2+^]_ex_ had the contrasting effect of initiating Ca^2+^ uptake, due to the increased driving force for the cation, while also promoting HC closure. This would account for the short-duration first peak. 

As for the delayed response producing the second peak (which was detectable at the final [Ca^2+^]_ex_ > 500 µM, Figure 4A), it is worth considering that keratinocytes (and several other cell types) sense changes in the [Ca^2+^]_ex_ via the G-protein coupled calcium-sensing receptor (CaSR) [90]. Activation of phospholipase C β isoforms downstream of CaSR-dependent G-protein stimulation results in the breakdown of phosphatidyl inositol-4,5-bisphosphate (PIP_2_), leading to the formation of inositol 1,4,5-trisphosphate (IP_3_) and diacylglycerol (DAG). DAG remains associated with the plasma membrane and activates protein kinase C, whereas IP_3_ mediates Ca^2+^ release from the endoplasmic reticulum (ER), by activating IP_3_ receptors in a ligand-dependent manner [91]. Note that IP_3_ receptors are co-regulated by Ca^2+^ that promotes their opening at low [Ca^2+^]_cyt_ and inhibits them at higher [Ca^2+^]_cyt_, depending on the IP_3_ level [92,93]. Therefore, the amplitude of this delayed, IP_3_-mediated response would depend strongly also on Ca^2+^ influx through HCs. In other words, the second peak that develops after CaCl_2_ bolus addition may be seen as and IP_3_-dependent (nonlinear) amplification mechanism for the signal generated by Ca^2+^ influx through HCs. This would explain its sensitivity to HC inhibitors in HaCaT-Cx26-GCaMP6s and HaCaT-Cx26G45E-GCaMP6s cells. The data in Figure 4A and Appendix A support this conclusion. However, dedicated experiments, beyond the scope of this article, are required to validate this hypothesis (e.g., siRNA-mediated knock down of the CaSR and/or pre-empting of the intracellular Ca^2+^ stores with the SERCA pump inhibitor thapsigargin). 

In the experiments with HaCaT-Cx46-GCaMP6s cells, Ca^2+^ uptake traces dropped below pre-stimulus levels in the recovery phase in the presence of Ab17396 at 0.1 µM and 1.0 µM but not with Ab17396 at 10 µM or with FFA at 100 µM. In contrast, in HaCaT-Cx26-GCaMP6s and HaCaT-Cx26G45E-GCaMP6s cells, the traces never dropped below baseline. Thus, we suspect that Ab17396 may affect, in a concentration-dependent manner, also other molecular players involved in the homeostasis of cytosolic Ca^2+^. In particular, low concentrations Ab17396 might facilitate the export of Ca^2+^ from the cytosol to the extracellular milieu, e.g., by potentiating the activity of plasma membrane Ca^2+^ ATP-ases or sodium–calcium exchangers. At higher concentrations, the prevailing effect of Ab17396 appears to be inhibition of Cx46 HCs. Further experiments are required to address this issue.

A stronghold of the Ca^2+^ uptake assay is the all-in-one Tet-on bicistronic system for the simultaneous, controlled expression of the Cx of interest and GCaMP6s in model cells. GCaMP6s provided not only an indirect fluorescent reporter for the successful expression of the Cx in HaCaT cells, but also a key element for the sensitive detection of [Ca^2+^]_cyt_ variations at the core of the method. Although we used LVs, other systems can be contemplated, including the piggyBac transposon Tet-on vector system [94]. 

Furthermore, the methodology can be readily extended to different Cx isoforms or pathological variants in other popular cell lines that express low levels of endogenous Cxs, e.g., HeLa cells [95]. At the opposite extreme, the assay could be used to probe native HCs in cells, organoids, native tissues (e.g., slices), or even *in vivo*. In all these cases, the requirements are (i) achieving adequate levels of the Ca^2+^ indicator (e.g., by *in vivo* viral transduction or electroporation) and (ii) being able to rapidly modify the [Ca^2+^]_ex_. Indeed, biasing (overexpressed or native) HCs towards the open state in a low [Ca^2+^]_ex_ medium prior to switching to a high [Ca^2+^]_ex_ medium to promote Ca^2+^ uptake is key for these experiments.

The strengths of this approach for the study of HCs can be summarized as follows. The all-in-one Tet-on system allows for inducible bicistronic expression of both the Cx isoform/mutant of interest and the biosensor required to test HC functionality (GCaMP6s). After Cx-expressing stable cell pools are established, no further manipulations are required to run the assay, ensuring homogeneity across samples. The assay can be run on large cell populations to rapidly obtain statistically robust results. It is sensitive enough to construct dose–inhibition curves for drug candidates, including mAbs, without the need for electrophysiological recordings of HC-mediated currents, which may be challenging in the case of leaky HCs. Last but not least, the assay can be extended to high-throughput high-content platforms and/or primary cells and microtissues.

Whereas dye uptake assays provide a useful complement, as also demonstrated here, they may lack the required sensitivity to discriminate among different experimental conditions (e.g., concentrations and/or inhibitors), chiefly due to charge/size of permeating fluorescent molecule in relation to HC pore size and electrostatic profile [96,97]. On the other hand, the time course of the fluorescence changes caused by dye accumulation is typically monotonic and thus comparatively easier to analyze and interpret.

## 4. Materials and Methods

### 4.1. Viruses

For controlled expression of GCaMP6s and the Cx of interest, we modified the pCW57.1 vector, a mammalian expression, lentiviral, Tet-on, destination vector with a puromycin selection marker (a gift from David Root; Addgene plasmid #41393; http://n2t.net/addgene:41393, accessed on 26 June 2022; RRID: Addgene_41393). A multiple cloning site (MCS) linked to an IRES and GCaMP6s sequences (MCS-IRES-GCaMP6s) was gene-synthesized and inserted between the NheI (GCTAGC) and BamHI (GGATCC) restriction enzyme sites in the pCW57.1 vector to form a new lentiviral transfer vector, which we named pCW57-MSC-IRES-GCaMP6s. 

Next, the MCS in this vector was used to subclone the CDS of human Cx26 (GJB2, NCBI CCDS ID: CCDS9290.1), its mutant Cx26G45E, or human Cx46 (GJA3, NCBI CCDS ID: CCDS9289.1). We thus obtained three corresponding inducible bicistronic lentiviral plasmids, named pCW57-Cx26-IRES-GCaMP6s, pCW57-Cx26G45E-IRES-GCaMP6s and pCW57-Cx46-IRES-GCaMP6s, respectively, for the simultaneous expression of the Cx of interest and cytosolic GCaMP6s. We deposited these plasmids in Addgene, where they have been respectively assigned the following IDs: 188236, 188237, 188238; https://www.addgene.org/Fabio_Mammano/, accessed on 26 June 2022.

### 4.2. Establishment of Stable Cell Pools

LVs were produced by transfection of HEK-293T cells (Cat. No. CRL-3216, American Type Culture Collection, ATCC, Manassas, VA, USA) using Lipofectamine 2000 (Cat. No. 11668-027, Thermo Fisher Scientific, TFS, Waltham, MA, USA) following a standard protocol [98]. LV-containing supernatant was collected, cleared by filtration and immediately used to infect HaCaT cells. The parental HaCaT cell line was maintained in Dulbecco’s Modified Eagle’s Medium (DMEM, Cat. No. 11995065, TFS) supplemented with heat-inactivated fetal bovine serum (FBS, 10% *v*/*v*, Cat. No. 10270-106, TFS) and penicillin/streptomycin solution (100 units/mL penicillin, 100 μg/mL streptomycin, Cat. No. 15070-063, TFS). 

The day before infection, 2.5 × 10^5^ cells were plated in a Petri dish (35 mm ∅) and cultured for 24 h. Then, the culture medium was removed and replaced with 0.5 mL of freshly collected undiluted LV-containing supernatant, supplemented with 8 µg/mL of hexadimethrine bromide (Cat. No. H9268, Merck KGaA, Darmstadt, Germany). The plate was incubated at 37 °C for 5 h with periodic gentle shaking every 5–10 min for the first 30 min; thereafter, 1.5 mL of normal culture medium was added, and incubation continued overnight. 

Twenty-four hours post infection, the LV-containing medium was replaced with fresh medium. Forty-eight hours post infection, 1 µg/mL of puromycin (Cat. No. P9620, Merck KGaA) was added to the medium and kept until a selected homogeneous cell pool was established (typically 3 days). The cell pools thus obtained were denoted as HaCaT-Cx(…)-GCaMP6s. For precise gene regulation of Tet-on pools, we replaced the ordinary FBS (see above) with Tet-approved FBS (Cat. No. A4736401, TFS). All cell pools were routinely tested for mycoplasma contamination by staining with DAPI (Cat. No. D1306, TFS) followed by visual inspection at the fluorescence microscope.

For imaging experiments or immunofluorescence analyses, 5 × 10^5^ cells were plated onto round glass coverslips (12 mm ∅) in a 35 mm ∅ dish two days before the assay. The next day and at least 24 h before experiments, dox (2 µg/mL, Cat. No. D1822, Merck KGaA) was added to the culture medium to promote Cx and GCaMP6s expression. To evaluate time-dependent dox-induced GCaMP6s expression in HaCaT-Cx26-GCaMP6s cells (Figure 1C), samples were washed and imaged with a spinning disk fluorescence microscope adapted for live-cell imaging [86] at different time points after dox administration (3, 6, 10, 24, 30 and 48 h). 

To estimate the percentage of pyknotic cells in HaCaT-Cx26-GCaMP6s and HaCaT-Cx26G45E-GCaMP6s pools (Table 2 and Table 3), cell-plated coverslips exposed to dox or vehicle for 24 h were fixed with 4% paraformaldehyde (PFA; Cat. No. 158127, Merck KGaA) for 15 min and permeabilized with 0.1% Triton X-100 (Cat. No. 1.08603, Merck KGaA) for 5 min. Next, samples were washed with PBS and nuclei were stained with DAPI (1 µM, 5 min). After visualization at the fluorescence microscope, cells that exhibited chromatin condensation and DNA fragmentation were counted in six different FOVs (*n* = 88–219 cells/FOV) for each condition, and the results were divided by the total number of cells in each FOV.

### 4.3. qPCR

The total RNA was extracted from HaCaT-Cx26-GCaMP6s and HaCaT-Cx26G45E-GCaMP6s cells exposed or not exposed to dox for 24 h using TRIzol Reagent (Cat. No. 15596018, TFS). 10 µg of the extracted total RNA was digested with DNase I (Cat. No. 18068015, TFS) and then purified using Rneasy mini Kit (Cat. No. 74104, Qiagen, Hilden, Germany) according to the manufacturer’s instructions. A NanoDrop spectrophotometer was used to quantify the RNA by measuring the absorbance and reverse transcription. (RT)-PCR was performed from 1μg of the purified RNA using high-capacity cDNA kit (Cat. No. 4368814, TFS), according to the manufacturer’s instructions but with a slight modification in which Oligo(dt) (Cat. No. 18418012, TFS) was combined with the cDNA kit random primers at a ratio of 1:5. 

Quantitative PCR (qPCR) was performed on the cDNA with specific primers (see below). Samples were analyzed in triplicate and gene expression levels were estimated by the 2^−∆∆CT^ method, using glyceraldehyde-3-phosphate dehydrogenase (GAPDH) as internal reference gene and Sybr green (SsoAdvanced Universal Sybr green Supermix, Cat. No. 1725274, BioRad, Hercules, CA, USA) on the ABI 7900HT sequence detection system equipped with the AB1 7900HT SDS software (Applied Biosystems, Waltham, MA, USA) applying the following amplification cycles:

50 °C: 2′, 

95 °C: 10′ 

95 °C: 0.15′, 65 °C: 0.35′ (40 cycles).

The primers used for these experiments are listed here.Cx26 (Cx26 wt and Cx26G45E) primers:

Forward 5′ATCGAAGGCTCCCTGTGGTG3′;

Reverse 5′ACAGTGTTGGGACAAGGCCA3′.

EMCV-IRES primers:

Forward 5′AATGTGAGGGCCCGGAAACC3′;

Reverse 5′ACTCACAACGTGGCACTGGG3′.

GCaMP6S primers:

Forward 5′ GGAGGACGGCAACATC 3′;

Reverse 5′GAAAGCCTCTTTAAATTCTGCG3′.

GAPDH primers:

Forward 5′CACCATCTTCCAGGAGCGAG 3′;

Reverse 5′CCTTCTCCATGGTGGTGAAGAC3′.

### 4.4. Immunofluorescence

Cells from stable pools grown on glass coverslips and exposed to dox or vehicle for 24 h were fixed with 4% PFA for 15 min, permeabilized with 0.1% Triton X-100 for 5 min and incubated for 1 h in blocking buffer containing 0.5% bovine serum albumin (BSA; Cat. No. A4503, Merck KGaA) and 0.1% Triton. Samples were then incubated overnight at 4 °C with anti-Cx26 polyclonal antibody (1:200, Cat. No. 71-0500, TFS) diluted in blocking buffer. 

The next day, samples were washed three times with PBS and incubated with cross-adsorbed Alexa Flour 555 secondary antibody (1:800, Cat. No. A-31572, TFS) for 3 h. For co-immunofluorescent staining, samples were then incubated overnight at 4 °C with anti-GFP mAb (1:50, Cat. No. A-11120, TFS) diluted in blocking buffer. The next day, samples were washed three times with PBS and incubated with cross-adsorbed Alexa Fluor 488 secondary antibody (1:800, Cat. No. A-21202, TFS) for 3 h.

For immunofluorescence labeling with abEC1.1h-IgG (Figure 4F), cells from stable pools, grown on glass coverslips and exposed to dox for 24 h, were incubated at 37 °C with ECM supplemented with 15 nM abEC1.1h-IgG for 1 h. Cells were then fixed with 4% PFA for 15 min and incubated for 1 h with a goat anti-human IgG Alexa Fluor 594 secondary antibody (1:800, Cat. No. 109-585-170, Jackson ImmunoReasearch, Ely, Cambridgeshire, UK). Samples were washed three times with PBS between all steps. 

Fluorescence images (pixel size 96 nm) were acquired with a confocal microscope (TCS SP5 Leica, Wetzlar, Germany) equipped with a 63× oil immersion objective (HC PL Apo, UV optimized, NA 1.4, Leica). Alexa Fluor 488 and GCaMP6s fluorescence was excited by a 488-nm Argon laser and collected between 500 nm and 540 nm. Alexa Fluor 555 and Alexa Fluor 594 fluorescence was excited by a 543-nm HeNe laser and collected between 568 nm and 680 nm and between 600 and 670 nm, respectively. For visualization purposes, a σ = 0.8 Gaussian filter was applied on the green channel of fluorescence images.

### 4.5. Dye Uptake 

DAPI (5 µM) was dissolved in ECM containing (in mM): 138 NaCl, 5 KCl, 0.06 CaCl_2_, 0.4 NaH_2_PO_4_, 6 D-Glucose, 10 HEPES (all from Merck KGaA), pH 7.3. For negative controls, ECM was supplemented with CaCl_2_ (2 mM) or flufenamic acid (FFA, 100 µM, Cat. No. F9005, Merck KGaA). Antibodies targeting the extracellular domain of HCs were dissolved in ECM at a concentration of 1 µM (abEC1.1m and abEC1.1h-IgG) or as stated in figure legends (anti-Cx46 antibodies).

For experiments with HaCaT-Cx26-GCaMP6s pools, coverslips with plated cells were washed with ECM and imaged using the two-photon microscope of Ref. [79] before and after 5 min incubation with one of the mentioned DAPI-containing solutions at room temperature. The microscope was equipped with a 25× water immersion multiphoton objective (XLPlanN25XWMP2, NA = 1.05, WD = 2 mm, Olympus Corporation, Tokyo, Japan). DAPI was excited at 780 nm and fluorescence emission signals were collected through a blue emission filter (460/50 nm, ET460/50m, Chroma Technology Corp, Bellow Falls, VT, USA). For each sample, we acquired multiple z-stacks (through-focus image sequences obtained with a 2 µm step size) of at least three different FOVs. FFA and abEC1.1m were pre-incubated at 37 °C for 30 min. 

For experiments with HaCaT-Cx46-GCaMP6s pools, the day before the experiment cells were seeded into a black 96-well microplate for high content imaging (2 × 10^4^ cells/well, #4580, Corning Inc., Corning, NY, USA) and dox (2 µg/mL) was added to the culture medium. The glass bottom of each well was pre-treated with laminin (2 µg/cm^2^ dissolved in water, 30 min at 37 °C; Cat. No. 11243217001, Merck KGaA). 

On the day of the experiment, cells were washed in ECM supplemented by 2 mM CaCl_2_ to remove FBS and mean GCaMP6s fluorescence value was measured at the plate reader (Varioskan LUX, Cat. No. N16045, TFS) for future normalization of DAPI uptake results to the number of cells present in each well. Then, wells were pre-incubated with one of the mentioned media for 20 min at 37 °C without DAPI (*n* = 3–6 wells/condition) to allow the blockers to act on HCs. Next, DAPI-deprived media were replaced with analogous DAPI-containing solutions and the microplate was moved to the plate reader to evaluate DAPI uptake rate. 

The mean DAPI fluorescence value for each well was measured every 10–15 min starting from 5 min after the onset of DAPI incubation (*t* = 0 in the graphs) and proceeded for 2 h. We ruled out any possible contribution of dead cells in the cultures (for which DAPI uptake is much faster than HC-dependent uptake in live cell) because for each well we treated the mean value of the first DAPI fluorescence reading (5 min after the onset of dye incubation) as a baseline and subtracted it from all the following measurements.

### 4.6. Ca^2+^ Uptake 

On the day of the experiment, cell-plated coverslips with confluent cell population were transferred to the stage of the spinning disk microscope described in Ref. [86] and equipped with water immersion objective (20×, N.A. = 0.95, XLUMPlan Fl, Olympus Corporation). Cells were incubated at room temperature for 15 min in ECM, thereafter fluorescence images were acquired at 1 frame/s for 12 min. GCaMP6s fluorescence was excited by a 488 nm diode laser (Cat. No. COMPACT-150G-488-SM, World Star Tech, Markham, Ontario, Canada) and emission signals were collected though a 535/30 nm band-pass filter (Cat. No. ET535/30M, Chroma Technology Corp., Bellows Falls, VT, USA). 

After 20 s of baseline recordings, a 2 µL bolus of 1 M CaCl_2_ solution was delivered through a manually operated pipette held at the edge of the FOV, and GCaMP6s fluorescence variations were monitored for 10 min. Finally, a 2 µL bolus of ionomycin solution (200 µM, Cat. No. I3909, Merck KGaA) was delivered to normalize the Ca^2+^ influx response to the maximum achievable fluorescence variation (i.e., to the level of biosensor expression throughout experiments or experimental sessions). We used each cell-plated coverslip only once. To assay the efficacy of HC inhibitors, coverslips were pre-incubated at 37 °C with ECM supplemented with one of the following HC compounds: 100 µM FFA (40 min), 1 µM abEC1.1m (30 min); abEC1.1h-IgG at a concentration varying from 0.001 nM to 10 µM for the construction of dose–inhibition curve (30 min). 

### 4.7. Selection of Anti-Cx46 mAbs

A synthetic, phage-displayed antibody library [99] was selected for binding to Cx46 protein expressed on the surface of Virus-Like Particles (Cx46-VLPs). Briefly, Expi293 cells (Cat. No. A14527, TFS) were co-transfected with plasmids expressing either HA-tagged Cx46 together with HIV Gag (Cx46 VLPs) or HIV Gag alone (empty VLPs), and VLPs were isolated from supernatant 2 days post transfection. Cx46 VLPs (100 µL at 10 µg/mL) were immobilized on Maxisorp Immuno plates (Cat. No. 12-565-135, TFS) and used for positive binding selections with library phage pools that were first exposed to similarly immobilized empty VLPs to deplete non-specific binders. After four rounds of binding selections, clonal phage was prepared and evaluated by phage ELISA (using Cx46 VLP or empty VLP) and sequencing as previously described [99].

### 4.8. Statistics

Statistical analysis was performed with MATLAB (R2019a, The MathWorks, Inc. Natick, Massachusetts, USA). The normality of distributions was assessed using the Shapiro–Wilk test. For normal distributions, statistical comparisons of means were made by ANOVA and post hoc comparison was by the Bonferroni test (multiple groups) or by two-tailed t-test (two groups); for non-normal distributions, statistical comparisons of means were made using the Kruskal–Wallis and post hoc comparison by the Dunn–Sidak test. The mean values are quoted ± s.e.m. unless noted otherwise. Sample sizes (n) of experimental groups are indicated in figure legends. *p* = *p*-values < 0.05 indicate statistical significance.

## Figures and Tables

**Figure 1 ijms-23-07337-f001:**
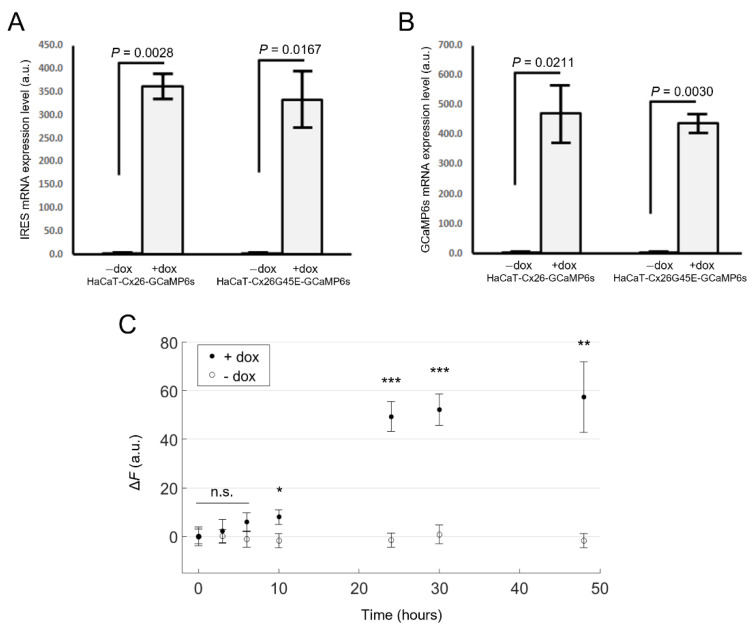
Validation of Tet-on bicistronic lentiviral constructs used to generate stable cell pools. mRNA expression levels of IRES (**A**) or GCaMP6s (**B**) assayed by qPCR (mean values ± standard deviation) in HaCaT-Cx26-GCaMP6s and HaCaT-Cx26G45E-GCaMP6s cells exposed to doxycycline for 24 h (+dox), or not exposed to doxycycline (−dox). *p*-values (*p*) were calculated using the two-tailed *t*-test. (**C**) GCaMP6s fluorescence levels in live HaCaT-Cx26-GCaMP6s cells vs. time after addition of dox (2 µg/mL, filled symbols) or vehicle (empty symbols) to the culture medium. The mean values ± standard error of the mean (s.e.m.) for ΔF = $F_{t}-F_{0}$ were computed for each condition as average fluorescence signal ($F_{t}$ ) of pixels that exceeded an arbitrary threshold minus the average value obtained before dox administration ($F_{0}$, *t* = 0). For each condition, values of *n* = 4–7 fields of view from one cell culture were averaged, and the statistical significance was calculated using a two-tailed *t*-test; *, *P* < 0.05; **, *P* < 0.01; ***, *P* < 0.001; and n.s., not significant.

**Figure 2 ijms-23-07337-f002:**
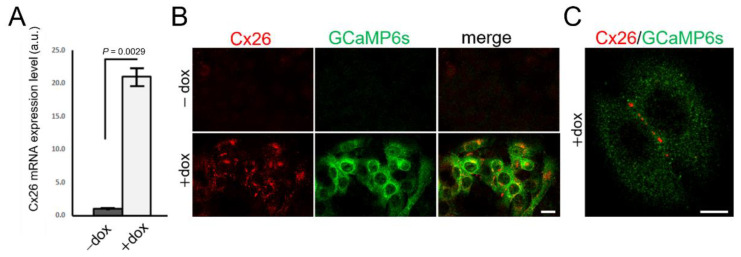
Characterization of Cx26 expression in HaCaT-Cx26-GCaMP6s cell pools. (**A**) qPCR assay of Cx26 mRNA expression levels (mean values ± standard deviation) in HaCaT-Cx26-GCaMP6s cells not exposed (−dox) or exposed (+dox) to dox (2 µg/mL, 24 h). *p*-values (*P*) were calculated using a two-tailed *t*-test. (**B**) Representative confocal images of fixed and permeabilized cells labelled with anti-Cx26 (red) and anti-GFP (green) antibodies (the GFP epitope recognized by the antibody is also present in GCaMP6s); scale bar: 20 µm. (**C**) Higher magnification view showing fluorescent puncta at points of contacts between adjacent cells; scale bar: 10 µm.

**Figure 3 ijms-23-07337-f003:**
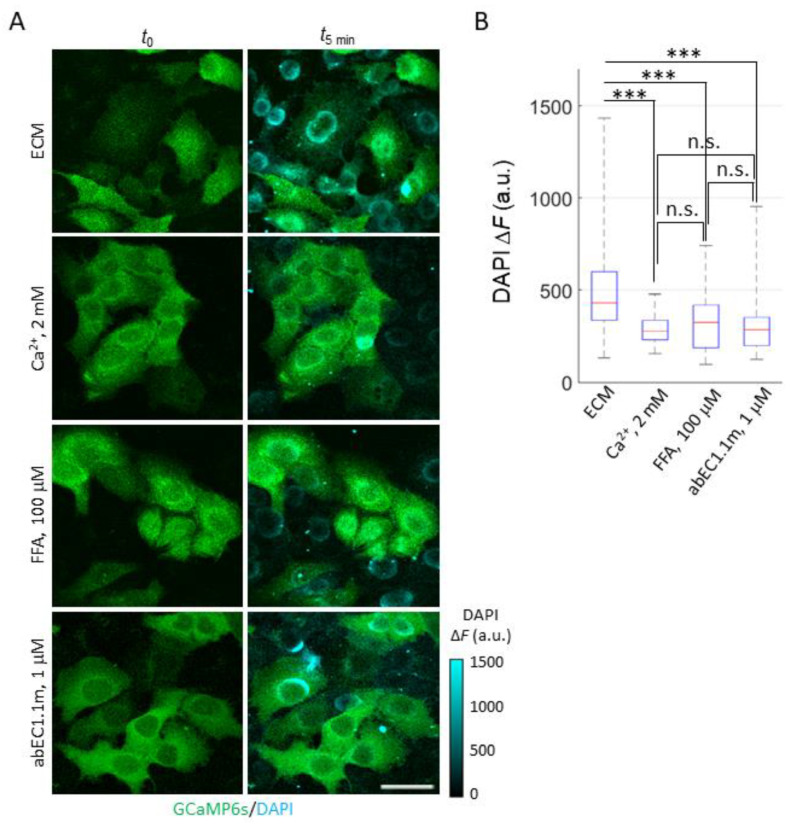
DAPI uptake in HaCaT-Cx26-GCaMP6s cells. (**A**) Fluorescence images were acquired using the two-photon microscope described in [79] before ($t_{0}$) and after 5 min of incubation ($t_{5\;\min}$ ) with DAPI (5 µM) in one of the following conditions (from top to bottom): ECM, ECM plus 2 mM Ca^2+^, ECM plus 100 µM FFA (denoted as FFA) or ECM plus 1 µM abEC1.1m (denoted as abEC1.1m). GCaMP6s signal (green) is shown as absolute fluorescence intensity (*F*). DAPI signal (cyan) is shown as ΔF = $F_{5\;\min}-F_{0}$. Scale bar: 40 µm. (**B**) Box plots of single cell ΔF distributions for DAPI uptake experiments described in (**A**); for each condition, DAPI uptake was measured in 45 ≤ *n* ≤ 80 cells. Red horizontal bars represent the median. Statistical significance was calculated using the Dunn–Sidak *post hoc* test for pairwise comparisons after the Kruskal–Wallis’ test; ***, *P* < 0.001; and n.s., not significant.

**Figure 4 ijms-23-07337-f004:**
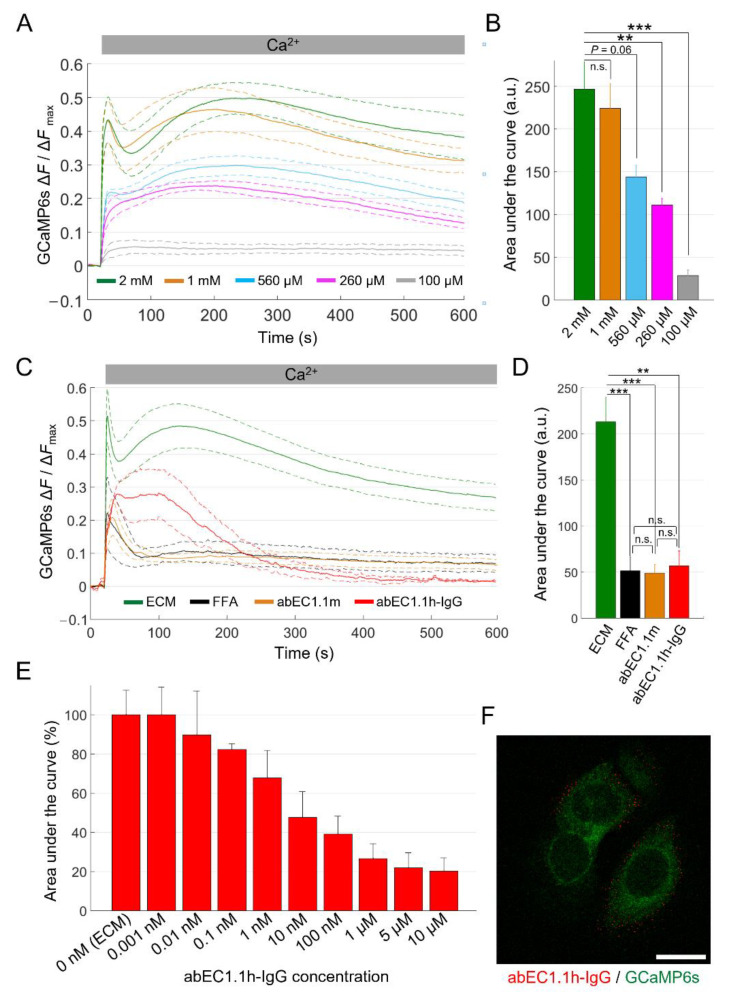
Ca^2+^ uptake in HaCaT-Cx26-GCaMP6s cells. (**A**) The mean Ca^2+^ uptake traces (solid lines) ± s.e.m. (dashed lines) obtained in HaCaT-Cx26-GCaMP6s cell cultures bathed in ECM and exposed to different CaCl_2_ bolus concentrations (BCs) to reach the following final [Ca^2+^]_ex_: 2 mM (green, *n* = 5, BC = 1 M), 1 mM (orange, *n* = 5, BC = 500 mM), 560 µM (light blue, *n* = 4, BC = 250 mM), 260 µM (magenta, *n* = 4, BC = 100 mM) and 100 µM (grey, *n* = 4, BC = 20 mM). Time *t* = 0 marks the onset of image acquisition. (**B**) Quantification of cytosolic Ca^2+^ load for the results in (**A**) computed as the areas subtended by the normalized Ca^2+^ uptake traces in the time interval between *t* = 20 s and *t* = 590 s (data are quoted as the mean ± s.e.m). Statistical significance was calculated using the Bonferroni *post hoc* test for pairwise comparisons after the ANOVA test; n.s., not significant; *P*, *p*-value; **, *P* < 0.01; ***, *P* < 0.001. (**C**) The mean Ca^2+^ uptake responses (solid lines) ± s.e.m. (dashed lines) to a 1 M CaCl_2_ bolus obtained from HaCaT-Cx26-GCaMP6s cell cultures bathed in the following extracellular media: ECM (green, *n* = 8), ECM plus FFA (100 µM, black, *n* = 4), ECM plus abEC1.1m (1 µM, orange, *n* = 7) and ECM plus abEC1.1h-IgG (1 µM, red, *n* = 3). (**D**) Quantification of cytosolic Ca^2+^ load for the results in (**C**) computed as described in (**B**). (**E**) Dose-dependent effect of the abEC1.1h-IgG for concentrations in the range 0.001 nM–10 µM (data are quoted as the mean ± s.e.m. for each condition). (**F**) Immunofluorescence labeling of HaCaT-Cx26-GCaMP6s cells with abEC1.1h-IgG; the antibody (15 nM) was applied to live cells that were then fixed but not permeabilized and counterstained with a red-fluorescent secondary antibody selective for the human Fc domain of the IgG. The green signal is due to GCaMP6s fluorescence. Scale bar: 20 μm.

**Figure 5 ijms-23-07337-f005:**
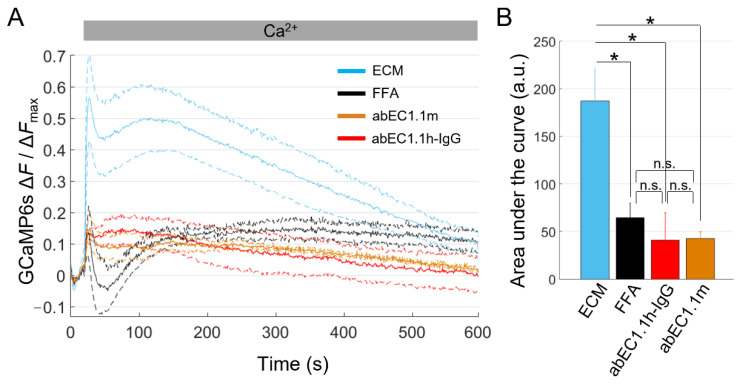
Ca^2+^ uptake through leaky Cx26G45E HCs is blocked by antibodies targeting the extracellular domain of Cx26. (**A**) The mean normalized Ca^2+^ uptake traces (solid lines) ± s.e.m. (dashed lines) obtained from dox-induced HaCaT-Cx26G45E-GCaMP6s cell cultures maintained in the following extracellular media: ECM (light blue, *n* = 7), ECM plus FFA (100 µM, black, *n* = 5), ECM plus abEC1.1h-IgG (1 µM, red, *n* = 4) or ECM plus abEC1.1m (1 µM, orange, *n* = 4). (**B**) The mean ± s.e.m areas subtended by normalized Ca^2+^ uptake traces in the time interval between *t* = 20 s and *t* = 590 s. Statistical significance was calculated using the Bonferroni *post hoc* test for pairwise comparisons after the ANOVA test; *, *P* < 0.05; and n.s., not significant.

**Figure 6 ijms-23-07337-f006:**
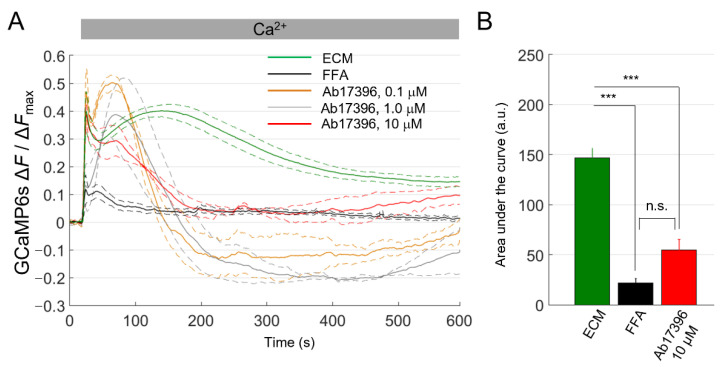
Ca^2+^ uptake assay in HaCaT-Cx46-GCaMP6s cells. (**A**) The mean normalized Ca^2+^ uptake traces (solid lines) ± s.e.m. (dashed lines) obtained from HaCaT-Cx46-GCaMP6s in ECM (green, *n* = 5), ECM plus FFA (100 µM, black, *n* = 6), ECM plus 0.1 µM anti-Cx46 mAb Ab17396 (orange, *n* = 3), ECM plus 1 µM Ab17396 (gray, *n* = 3) or ECM plus 10 µM Ab17396 (red, *n* = 3). (**B**) Areas subtended by normalized Ca^2+^ uptake traces (mean ± s.e.m.) in the time interval between *t* = 20 s and *t* = 590 s for some of the data shown on the left; 0.1 µM Ab17396 and 1 µM Ab17396 conditions were omitted due to the negative swing; this was never observed with other inhibitors or in control conditions. Statistical significance was calculated using the Bonferroni *post hoc* test for pairwise comparisons after the ANOVA test; ***, *P* < 0.001; n.s. not significant.

**Figure 7 ijms-23-07337-f007:**
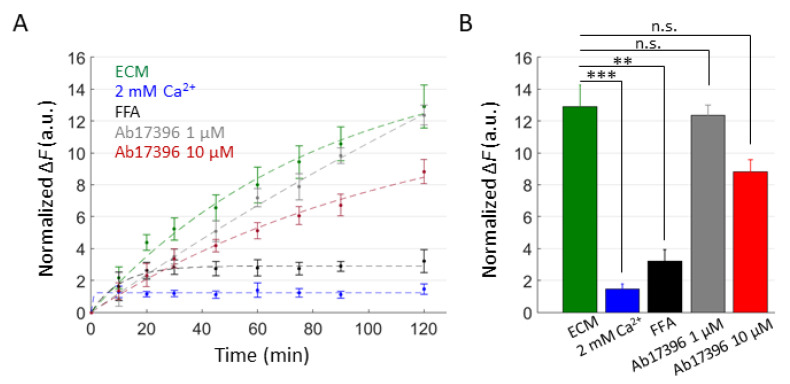
DAPI uptake assay in HaCaT-Cx46-GCaMP6s cells. (**A**) Time course of DAPI uptake assay performed with a plate reader in dox-induced HaCaT-Cx46-GCaMP6s stable pools. Cells were exposed to DAPI (5 µM) dissolved in: ECM (green), ECM supplemented with 2 mM Ca^2+^ (blue), ECM plus FFA (100 µM, black), ECM plus Ab17396 1 µM (gray) and ECM plus Ab17396 10 µM (red). In each condition, DAPI uptake was measured as the average fluorescence signal variation ΔF = $F\left(t\right)-F_{0}$ of 3 ≤ *n* ≤ 6 cultures at different time points, from 5 min after the beginning of dye incubation (*t* = 0) up to 2 h later. The data are quoted as the mean values ± s.e.m. (**B**) The mean ΔF values obtained for each condition shown in (**A**) at the end of the observation time window (2 h); statistical significance was calculated using the two-tailed *t*-test; **, *P* < 0.01; ***, *P* < 0.001; and n.s., not significant.

**Figure 8 ijms-23-07337-f008:**
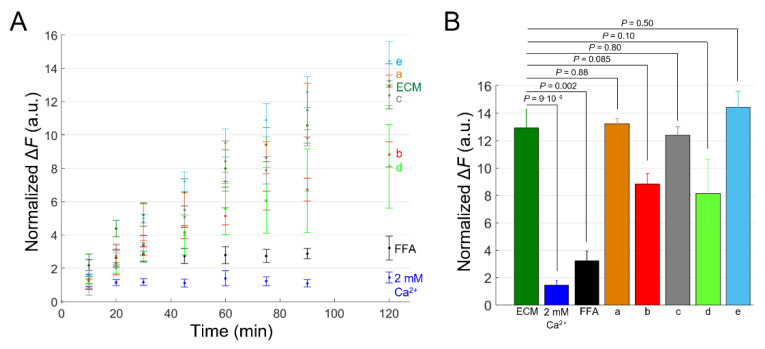
Tests of different anti-Cx46 candidate mAbs by DAPI uptake assay in HaCaT-Cx46-GCaMP6s cells. (**A**) Time course of DAPI uptake assay performed with a plate reader in dox-induced HaCaT-Cx46-GCaMP6s stable pools. Cells were exposed to DAPI (5 µM) dissolved in: ECM (green), ECM supplemented with 2 mM Ca^2+^ (blue), ECM plus FFA (100 µM, black), ECM plus Ab17340 (a, 1 µM, orange), ECM plus Ab17396 (b, 10 µM, red), ECM plus Ab17397 (c, 1 µM, gray), ECM plus Ab17398 (d, 0.4 µM, light green) and ECM plus Ab17399 (e, 1 µM, light blue); for each condition, DAPI uptake was measured as average fluorescence signal variation ΔF = $F\left(t\right)-F_{0}$ of 3 ≤ *n* ≤ 6 cultures at different time points, from 5 min after the beginning of dye incubation (*t* = 0) up to 2 h later. The data are quoted as the mean values ± s.e.m. (**B**) The mean ΔF values obtained for each condition shown in (**A**) at the end of the observation time window (2 h); *p* values (*P*) were calculated using the two-tailed *t*-test.

**Table 1 ijms-23-07337-t001:** Rare genodermatoses caused by leaky/hyperactive HCs. (P = prevalence; C = number of cases described in the literature; and E = estimated number of people with this disease).

Type of Disease (Acronym/OMIM ^§^ No.)	P	C	E	Cx (Gene)	Amino Acid Mutations [62]
Keratitis-ichthyosis-deafness syndrome, autosomal dominant (KID/148210)	<1/1,000,000 ^†^,	<100 ^†^	1–300 ^╬^	Cx26 (*GJB2*)	G12R, N14K, N14Y, S17F, I30N, A40V, G45E, D50A, D50N, D50Y, A88V
Palmoplantar keratoderma with deafness (PPKDFN/148350)	<1/1,000,000 ^†^,	5 ^§^	1–300^╬^		H73R, S183F
Ectodermal Dysplasia 2, Clouston type (ECTD2/129500)	1–9/100,000 ^†^,	>150 ^‡^	3000–30,000 ^╬^	Cx30 (*GJB6*)	G11R, A88V
Palmoplantar keratoderma and congenital alopecia 1 (PPKCA1/104100)	<1/1,000,000 ^†^,	10 ^†^	NA ^#^	Cx43 (*GJA1*)	G8V
Erythrokeratodermia variabilis et progressiva 3 (EKVP3/617525)	<1/2,000,000 ^†^	3 ^§^	NA ^#^		A44V, E227D

**^§^** Online Mendelian Inheritance in Man (OMIM), a catalog of Human Genes and Genetic Disorders (www.omim.org) accessed on 26 June 2022; ^†^ According to the Orphanet database (www.orpha.net) accessed on 26 June 2022; ^‡^ According to GeneReviews^®^ [Internet] (www.ncbi.nlm.nih.gov/books/NBK1200/) accessed on 26 June 2022; ^╬^ in the U.S.A., according to the Information Center on Genetic and Rare Diseases (GARD) database managed by the NIH (https://rarediseases.info.nih.gov/) accessed on 26 June 2022; ^#^ Data not available; the disease is not listed on GARD but is present in Orphanet, OMIM and the MalaCards databases.

**Table 2 ijms-23-07337-t002:** Percentage of pyknotic nuclei in HaCaT cell pools overexpressing Cx26 or Cx26G45E, exposed (+dox) or not exposed to dox (−dox) for 24 h.

Cell Pool	Average Percentage
Cx26 −dox	1.86 ± 0.21
Cx26 +dox	3.4 ± 0.15
Cx26G45E −dox	3.11 ± 0.23
Cx26G45E +dox	6.65 ± 0.86

**Table 3 ijms-23-07337-t003:** Statistical comparisons for results presented in Table 2 calculated with the Bonferroni *post hoc* test for pairwise comparisons after the ANOVA test.

Cell Pools	*p*-Value
Cx26 −dox vs. Cx26 +dox	0.7703
Cx26G45E −dox vs. Cx26G45E +dox	0.1574
Cx26 −dox vs. Cx26G45E −dox	0.8642
Cx26 +dox vs. Cx26G45E +dox	0.2172
Cx26 −dox vs. Cx26G45E +dox	0.0346
Cx26 +dox vs. Cx26G45E −dox	0.9975

## Data Availability

Data were archived in a repository handled by the University of Padova and can be accessed at: http://researchdata.cab.unipd.it/645/, accessed on 26 June 2022. Lentiviral plasmids developed and used in this study have been deposited and will be made publicly available via the Addgene repository with the following IDs: 188236, 188237, 188238; https://www.addgene.org/Fabio_Mammano/, accessed on 26 June 2022.

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
