# Peer review of "A Quantitative Assay for Ca^2+^ Uptake through Normal and Pathological Hemichannels"

_ijms, 2022, doi:10.3390/ijms23137337_

Round 1
Reviewer 1 Report
Connexin (Cx) hemichannels (HCs) are plasma membrane channels permeate to ions, metabolites, and a variety of other molecules including Ca2+. HC inhibitors are attracting growing interest as drug candidates. Nardin et al. developed an all-optical assay, based on fluorometric measurements of Ca2+ uptake with a Ca2+-selective genetically encoded indicator GCaMP6s, to determine the effect of HCs inhitor on HC activity. The method might be useful in screening monoclonal antibodies (mAb) targeting the HCs for diseases therapy. Overall, the manuscipt is well presented, however, I have the following concerns:
1. For Fig. 3A, The microscope images of DAPI should also be given for the results of DAPI uptake in HaCaT-Cx26-GCaMP6s cells inhibited by flufenamic acid and abEC1.1m. In addtion, whether various Ca2+ concentration would influence the expression of Cx26? For this purpose, the Cx26 expression detected using anti-Cx26 antibody might be given.
2. The methods described for experiments in Figs. 4A and B would be suitable if the aim was to evaluate the function of GCaMP6s protein as an intracellular Ca2+ indicator. But if GCaMP6s protein is used as an indicator of Cx26 transporting function, then Ca2+ should be added as less than 2 mM, possibly 0.06 mM, as high concentration Ca2+ is supposed to close the Cx26 channel. In additon, why the flurescence intensitiy of HaCaT-Cx26-GCaMP6s cells at 600S was lesser than that in the previous time points, such as 240S?
3. For Figs. 4D and 4E, the bright field and microscopy images should be given for the effect of FFA, abEC1.1m and abEC1.1h-IgG addition on the Cxs channel activity.
Reviewer 2 Report
This paper reports a quantitative assay for Ca2+ uptake through connexin hemichannels. The authors designed a GCaMP6-based fluorescence measurement of Ca2+ uptake. This assay proved to be very useful in screening for inhibitors of connexin hemichannels. The paper is well written and easy to understand. Just one small suggestion.
Fura-2 dye is frequently used for intracellular Ca2+ measurements. If Fura-2 based Ca2+ measurements can be performed to obtain similar results (especially in the panel of Figure 4C) to this new GcaMP6 method, this will provide cross-validation results for this new method.
